# Breast Cancer Survivors’ Motivation to Participate in a Tailored Physical and Psychological Intervention: A Qualitative Thematic Analysis

**DOI:** 10.3390/bs12080271

**Published:** 2022-08-07

**Authors:** Valeria Sebri, Ilaria Durosini, Davide Mazzoni, Gabriella Pravettoni

**Affiliations:** 1Department of Oncology and Hemato-Oncology, University of Milan, 20122 Milan, Italy; 2Applied Research Division for Cognitive and Psychological Science, IEO, European Institute of Oncology IRCCS, 20122 Milan, Italy

**Keywords:** breast cancer survivors, physical activities, psychological support, integrated interventions

## Abstract

*Purpose:* Participants’ engagement in a project requires intrinsic motivations, which may evolve during the intervention thanks to lifestyle changes and positive challenges. Over the years, tailored programs based on physical activity and psychological sessions have been promoted to support the quality of life and well-being of breast cancer survivors. Personal expectations and needs are essential to predict participant adherence to the intervention as well as their possibility to reach positive outcomes. This study presents a preliminary understanding of the differences between motives and outcomes obtained after an integrated physical and psychological intervention conducted by professional trainers and psychologists. *Methods:* Forty-five women with a history of breast cancer answered some questions before and after the program, and the results were analyzed in accordance with the procedure of the thematic analysis. *Results:* Physical and psychological well-being are the two main themes that emerged from the participants. Interestingly, some differences emerged between the two data collections. Themes such as aesthetic evaluation interest and the need to learn psychological strategies disappeared at the end of the intervention; on the other hand, the need to make a distance from the illness experiences emerged as an obtained outcome. *Conclusions:* The discussion explains these differences and highlights the importance of considering breast cancer patients’ needs and motives to take part in interventions to promote quality of life.

## 1. Introduction

Receiving a diagnosis of breast cancer exposes people to physical and psychological consequences that persist over time and affect their quality of life. A patient’s path does not end with successful physical care but continues even after treatment. In this regard, people have to deal with negative emotions related to the diagnosis (such as emotional distress, anxiety, and fatigue), changes in lifestyle and social relationships, and possible long-term side effects of oncological care on survivors’ health (e.g., fatigue, vomiting, pain, and risk of infections) [1,2]. In addition, breast cancer could be disruptive to a woman’s identity. For example, it may affect the body (e.g., breasts) related to a woman’s expression of femininity or put directly at risk the possibility of a pregnancy. In addition, the mental representation of one’s own body and the related emotions within an overall sense of bodily self (i.e., body image; [3,4] could be damaged after the diagnosis and treatments).

Thus, it is of paramount importance to support breast cancer patients, helping them to overcome the emotional trauma related to the diagnosis, manage their emotions, and re-define their life after cancer [5,6]. In this sense, a growing body of literature evidences the efficacy of psychological interventions for breast cancer patients and survivors to promote a deeper understanding of the psychological inner world and improve personal well-being. For example, group psychological interventions may encourage breast cancer patients to explore their life experiences in a safe context with psychotherapists and group members that have similar life experiences. In this regard, group psychotherapy could be useful to enhance the sharing of non-metastatic patients’ disease-related emotions and promote a better quality of life [7]. Furthermore, literature shows that short integrated interventions combining group psychotherapy programs and physical activity are effectively used with breast cancer survivors to promote beneficial effects on personal strength, cognitive abilities, and new social bonds [8,9,10]. Sport performance helps patients to strengthen cognitive and functional skills [11]. Notable introspection could be promoted through the discovery of personal abilities during physical activities, helping people to build a new sense of personal self after the diagnosis [12], improving their capacity to transfer such abilities in general life (e.g., improving cancer management); [4] and fostering social support and connections [8]. Recent meta-analyses and reviews have highlighted that sport-based programs have positive effects on the health management and quality of life of breast cancer survivors [13], although it has been demonstrated that some sports are less effective than others in the literature [14].

In Italy, some integrated interventions based on physical activities and psychological support to improve quality of life in cancer survivors were promoted [8]. In the last few years, two new projects to support women who have lived with breast cancer have been promoted to help the women obtain a new self-representation, resources, and abilities. In both of these projects, women who received oncological diagnoses were invited to take part in physical and psychological group intervention programs. Sailing activities or postural exercise sessions were promoted by professional trainers and were coupled with psychological support group sessions conducted by psychologists.

The psychological literature highlights that the involvement of cancer survivors in interventions often requires intrinsic motivation, which often appears to have a dynamic nature [8,15]. Motives usually refer to a range of outcomes or states that individuals aim to attain or avoid [16] and are inherent within several theories, like social cognitive theory [17]. Rather than being stable, motives that change over time are influenced by personal and social experiences. As suggested by recent research [8], social experiences characterized by novelty, group engagement, peer influence, and active mentoring promote perceived cohesion, self-efficacy, and lower negative emotions (such as depression and anxiety). Moreover, changes in motivations over the course of interventions can impact personal needs and emotional dynamics, which can lead people to reach different goals compared to initial expectations [8].

On this basis, the present study aims to qualitatively explore the initial motivations and perceived outcomes in female cancer survivors’ participation in two different programs that combine sports activities and group psychological support to promote quality of life. In the context of the above-mentioned integrated intervention, a qualitative thematic analysis was carried out to assess the motivations and outcomes of participants who took part in the intervention. The analyses highlighted the main themes (and sub-themes) of internal motivation that participants showed at the beginning of the intervention and the perceived outcomes that participants reported at the end of the project.

## 2. Materials and Methods

### 2.1. Participants

A total of 45 women with an oncological history were invited to take part in this study (*M_age_* = 50.24; *SD_age_* = 6.75). All the participants included in this study met the following inclusion criteria: (1) adult participants (2) women who received a diagnosis of breast cancer in the past few years, and (3) participants that were not actually under oncological treatment. Women who showed cognitive impairment, physical or psychological impairments that prohibited their participation in the study, or an inability to understand the study or to sign the informed consent were excluded from this research.

The majority of participants had attained a high school education (59.9%), lived in the North of Italy (74.5%), and worked as white-collar employees (92.7%). Additionally, more than half of the sample participates in physical activities (60.5%) and is not involved in individual psychotherapy (63.6%) (see Table 1).

Participants voluntarily took part in an intervention based on physical exercises and a psychological program that aims to support well-being and quality of life through physical and psychological sessions with an expert psychologist.

The sample was constituted as follows: 26 women (*M_age_* = 50.13; *SD_age_* = 8.00) participated in a group intervention that combined physical/postural exercises with psychological group interventions for a total of 13 sessions (nine sessions of physical activities and four sessions of group psychotherapy); 19 women (*M_age_* = 50.32; *SD_age_* = 5.65) participated in a group intervention coupling sailing boat courses with psychological group interventions for a total of eight sessions (eight sessions of physical activities and seven sessions of group psychotherapy).

All the trainers involved in these interventions have extensive professional experience in postural exercises and sailing boat courses. The physical intervention sessions (both sailing and postural exercise courses) were tailored on the basis of participants’ needs and abilities. In particular, physical activities lead to the promotion of a positive body image [18], emotional regulation (for example, regulation of anxiety, depression, and distress), cognitive development (e.g., attention, memory, and decision-making processing) [11,14], and individual self-esteem [19]. Improvements in muscle tone and weight allowed patients to be more willing to wear clothes that were more revealing, feel better about themselves, and develop a strong relationship with their loved ones, thanks to their high self-confidence [20]. Additionally, all the participants received 2 h of group psychotherapy from expert psycho-oncologists in order to explore their personal goals within the intervention, manage their emotions, and share their illness experiences in a group. Specifically, psychological interventions aimed at decreasing emotional distress and negative behaviors that are relevant to sustain the adjustment to the illness experience and avoid self-fragmentation [2]. Accordingly, psychological sessions were based on studies that decreased participants’ distress by providing problem-solving methods and relaxation techniques to modify the perception of one’s own body [21].

### 2.2. Procedure

The invitation to participate in the programs was posted on social networks and sent via a commercial mailing list. Breast cancer survivors who accepted to participate received the information sheet and informed consent form. At the beginning and end of the intervention, the participants were invited to answer some open questions. Specifically, at the beginning of the program, participants described their initial motivations to participate in the program with the help of some stimulus questions (i.e., What prompted you to decide to participate in this project? What aspects did you take into consideration when deciding to participate in this project? What goals did you set yourself by participating in this project?). At the end of the program, the same participants described the main outcomes they obtained thanks to the participation in the project (i.e., Have you achieved your objectives related to your participation in this project? What are the main objectives achieved by participating in this project?). All the textual answers were collected through Qualtrics software. Participants responded to the questions individually; no time or word-limit constraints were given.

### 2.3. Corpus Preparation

All the text answers provided by participants were downloaded from Qualtrics as a .xls file. All the reports were in Italian, and all the references to people, locations, and dates were anonymized. The length of the answers was heterogeneous, ranging from a few-word statement to a paragraph of 280 words for motives and 223 words for outcomes.

### 2.4. Data Analysis

The stages of the analytical process followed the main phases of qualitative thematic analysis, as described by Braun and Clarke [22]. Three researchers analyzed the data independently (V.S., I.D., and D.M.). A qualitative thematic analysis with a bottom-up approach was used, focusing the attention on the themes emerging from the data rather than trying to fit the data into a pre-existing coding frame [23]. Specifically, the analytical approach of this study was mostly “semantic” [24] and themes were identified within the explicit meanings of the data (not looking for anything beyond what our participants wrote). According to Braun and Clarke [22], we followed these five steps:In the first phase, two authors (V.S. and D.M.) read each text many times, familiarizing themselves with the content.In the second phase, authors conducted an initial coding of the data; codes were used to identify segments of the textual reports (semantic content). If one answer reported a single main motive or outcome, it was considered a single segment. This initial coding procedure was run with Microsoft Excel.In the third phase, the different codes were grouped into potential main themes (and sub-themes). Discrepancies between the raters were resolved through a discussion between the authors.In the fourth phase, we developed and reviewed the themes. Specifically, the thematic map was reviewed by two authors (V.S. and D.M.).In the final phase, themes were labeled, providing an explicit definition of their themes. To further validate the thematic map, one author who did not take part in the previous phases of analysis (I.D.) reviewed the entire process and the identified themes.

The authors had several meetings to discuss and define the themes. Five main themes were identified, with 12 sub-themes for motivations and 10 sub-themes for outcomes.

## 3. Results

The results of the analytical process are presented in this section. Five main themes, 12 sub-themes for motivations, and 10 sub-themes for outcomes were identified (Table 2). According to Hannah and Lautsch [25], results were presented using a non-numerical form. Women’s quotations were coded using participants’ ID numbers.

### 3.1. Physical Well-Being

The first main theme is physical well-being. Women included in this study highlighted the need to promote physical well-being through their participation in the integrated intervention sessions.

Specifically, before starting the intervention, some breast cancer survivors reported their need to do physical activities to *promote their physical well-being and their relaxation*. For example, a woman who received, in the past, a breast cancer diagnosis stated that she participated in this intervention in order to: “improve my well-being and do physical activities in order to avoid incorrect body postures (ID11)”. Similarly, other participants declared that their aims were to: “lengthen and stretch the body to reduce physical pains and lymphedema (ID12)” and “decrease the bodily tensions (ID25)”. Additionally, another sub-theme was the promotion of *body awareness*. Women reported their desire to stay in contact with their body and related inner sensations in order to improve their self-awareness (e.g., “My goal is to regain contact with my body by relying on a physical trainer who will suggest me harmless exercises, taking into account my physical difficulties (ID19)” and “Tone my muscles and be aware of my body. I need to know my bodily sensations (ID6)”). Lastly, some women included in the project reported that their initial motivation to participate was to increase their *aesthetic evaluation* in order to improve their self-acceptance (e.g., “Find harmony with my body. I have always been comfortable with my esthetical appearance, now I do not feel myself good (ID2)”).

After engaging in the intervention, breast cancer survivors reported improvements in their physical well-being. From the words of some participants, it emerged that the integrated intervention helped women to promote their *body sensations and perceptions*. For example, a woman stated that: “I feel well, the intervention sessions were a support and an effective method of relaxation (ID1)”. Moreover, several women perceived lower *body tension and more relaxation* (e.g., “Meet other people with an oncological history was very useful for me. It was very helpful to do gymnastics and try to relax after a long day. I don’t know if I had any advantages for my migraine disease, but I’ve been pretty well these weeks (ID4)” and “I perceive more relax related to neck; I will continue with physical exercises twice a week before sleeping (ID8)”. At the same time, participants evidenced improvements in their *body awareness*, in terms of available physical movements, as reported: “I became more aware of the possibilities of making some movements, also trying to make physical movements that I had not done in a long time. Moreover, the comparison with the other women of the group helped me to reflect on aspects of myself that I have not ever explored before (ID6)” and “Participating in this project, I learned to be aware of body parts that I felt in tension (ID14)”. We also noticed that few breast cancer survivors demonstrated other physical benefits, which are not associated with the intervention directly, for example: “Thanks to this intervention, I learned to relax some sore parts of my body, improving my quality of sleep (ID14)”. This highlights the possibility of reaching other goals that were not previously set at the beginning of the intervention. Moreover, related to the differences between the initial motivations to participate in the program and the final outcomes, we noticed that the *aesthetic aspect* theme disappeared.

### 3.2. Psychological Well-Being

Psychological well-being is the second main theme reported by participants. One of the main motivations for being engaged in the integrated intervention was focused on achieving *self-awareness* of their inner world in order to improve their understanding of emotions, thoughts, and sensations. For instance, women with a history of breast cancer reported their desires to promote psychological abilities to “try to give a meaning and put a name to my feelings (ID14)”. In line with this, a women reported: “I chose to participate in this program because I would like to have more confidence in myself. Additionally, I want to be aware and recognize my fragility, have the opportunity to share my thoughts and experiences, and to promote healthy habits to appreciate myself and my life (ID14)”. These motives were strictly associated with participants’ desires of promoting psychological abilities to cope with negative emotions, as reported: “Understanding my anxiety and anger that are always present in my life (ID25)” and “Gaining more awareness of myself, of my resources and my limitations; in other words, improving and accepting myself (ID20)” in order to promote “A greater acceptance of myself, the desire to see me with different eyes and react to a difficult period of my life (ID24)”. Additionally, participants highlighted their initial motivation to take part in the intervention to *have time to make new experiences*, far from their usual everyday life. For example, women reported motivations linked to the desire to “Stay with myself and challenge myself (ID34)” and “Take time for myself. Develop a better awareness of my limits and abilities and do what I like (ID40)”.

Referring to outcomes after interventions, breast cancer survivors demonstrated a general satisfaction about their psychological well-being. Participants reported better *self-awareness*, reporting to obtain “More awareness and serenity because I have worked on myself by facing some negative thoughts about myself (ID24)” and “Now, I feel myself free from a burden. I have less anxieties and useless thoughts. I feel I have new abilities to deal with some of my difficulties and stay in contact with others. I have learned to better understand myself and my emotions (ID35).” In line with initial motivations, some participants reported that this intervention helped them to both overcome *negative emotions* (e.g., “I have overcome so many of my fears. I have let go the need to control everything and I have rested my mind. This has been possible thanks to the psychologist and the group of women with whom I shared this wonderful experience (ID41)” and *take time for themselves* (“I have better confidence in myself and my abilities. I learned to see myself from the outside by not affecting my judgment with limiting thoughts. I learned to take time for myself and meditate on my life (ID16)”).

### 3.3. Coping with the Illness

Participants’ motivations were also related to their needs to *overcome the illness experience* and improve their ability to turn negative feelings into positive growth. A participant wrote: “After the second diagnosis of cancer, I could not elaborate my illness because I experienced fear and anger. I decided to participate instinctively because I felt the need to do something for myself (ID39)” and “I want to live a unique experience, thinking about my illness in a positive way and comparing myself with other people who received my same diagnosis. Talking about my illness, listen other experiences, and take a break from my daily routine could be useful for me…I think that staying away from my daily routine could help me to understand who I am and what I feel (ID41)”. Similarly, other participants evidenced *physical and emotional challenges* (“I want to achieve a positive well-being, accepting my mastectomy. I want to accept my changed body: my reduced energy, my 15 extra pounds, my changes in metabolism.... Sometimes, I am not comfortable with the expression of my negative emotions and feelings (such as sadness or anxiety) because I think that people who received a worse diagnosis than me (e.g., women with metastatic breast cancer) have to face strong difficulties. At the same time, I know that the expression of my emotions can help me (ID9)”). In addition, they directly expressed the desire of *learning psychological strategies* to face everyday life difficulties, as reported in the following quotations: “My goal is to improve my self-esteem and, in particular, acquire useful “tools” to accept my illness and my body changes. I realize that my self-esteem sometimes falters, sometimes I feel discouragement, especially when pain and edema in the arms intensified, or when I wake up tired. These moods affect my well-being (ID9)”.

After the intervention, the main outcome reported was related to improvements in the overall illness acceptance, in line with the motives. Breast cancer survivors evidenced improvements in *self-awareness* and skills to face physical and psychological issues (e.g., “After the intervention, I have greater awareness of my emotions. Take part to a group of women who had breast cancer and share with them feelings, thoughts, and emotions lead me to be aware of my and others suffering (ID44)”. Another interesting outcome referring to coping with the illness abilities is the new perception of being a woman, and not only a patient. This is relevant in terms of self-representations; in other words, breast cancer survivors evidenced a new focus on themselves as people, with needs, desires, and goals. In particular, they put themselves at the center of their lives (e.g., “I started physical activities after a long time of a sedentary period. I put myself and my dreams at the center of my life (ID37)”). Participants also reported the desire to distance themselves from the idea of being an ill person (e.g., “After this intervention, I take distance to my oncological disease. I have the possibility to learn something new from this negative illness experience and I am less worry about my negative emotions (ID9)”).

### 3.4. Social Relationships

Positive social relationships emerged as a relevant point of interest as one of the motives to participate, thanks to the possibility of perceiving group belongingness. More specifically, breast cancer survivors reported the need of *sharing their illness experience* (e.g., “I decided to participate in this intervention to share my thoughts and emotions, freely. I thought it could be a relevant opportunity for my growth, promoting new abilities and allowing a positive comparison with other women with a history of breast cancer (ID22)” and “I have the desire to share my experience with other women, gain more awareness of myself, and take time to my emotional and physical growth (ID20)”) and listened to others (e.g, “I want to meet new people, listen to their life stories, and share my illness experiences with them (ID42)”). An additional important initial motivation to take part in the intervention was related to the need to *stay in contact with people who lived a similar illness experience* (“For me, it is very important to share my emotions and feelings with women who lived a similar illness experience (ID6)”). Finally, having a *previous positive experience in a group* intervention was reported as motivation to participate in a new project. A woman indeed wrote: “I decided to participate in this intervention because I would like to experience positive relationships as I lived in other similar projects (ID12)”.

In line with motivations, *sharing experience* and emotions with others was one of the main reported outcomes. Breast cancer survivors evidenced a general satisfaction, referring to the possibility of discussing their illness with other cancer survivors to reflect on their experiences. This way, a woman wrote: “The discussion with other women in the group helped me to reflect deeply about myself and my body after illness. Moreover, I had the occasion to discuss the relationship with my partner and our intimacy issues (ID6)” and “I shared a lot of my emotions during this intervention and I understand that I can do it! I did not feel sick. Listening to others and sharing thoughts and emotions enriched my point of view. Feeling myself surrounded by new friendships was very useful and I will carry on forever with me. I was authentic and I was myself during this intervention; and now… I feel free”! (ID39)”. Additionally, another outcome is related to the relevance of the group to *improve yourself and personal habits.* “During this intervention, the main result is to be out of my daily routine in which I felt myself trapped. The discussion and comparison with women involved in this project lead me to change of habits, improving physical activities especially. My body will never come back like before; however, I can learn to feel myself good in my new body, without forgetting to take care of my inner world. I am the only one responsible for my well-being and body perception (ID9)”.

### 3.5. Support to Research

Interestingly, some breast cancer survivors also reported the need to be useful to improve scientific studies in the psycho-oncological field. In particular, some of them wrote the following sentences: “My goal is to give a little contribution to psycho-oncology research (ID21)”. This is associated with the interest in promoting helpful studies for future breast cancer survivors (e.g., “I believe in the importance of these studies and I hope that my little contribution will be useful for me and for women who lived similar illnesses (ID10)” and “Trying to develop studies to help people who will be affected by breast cancer (ID16)”. Accordingly, breast cancer survivors also demonstrated their satisfaction of having supported scientific studies as an outcome (“My goal was to provide data for this study (ID43)”.

## 4. Discussion

The present paper aimed to describe the main motives and perceived outcomes related to the participation in an integrated intervention to promote well-being in a group of breast cancer survivors. On a theoretical level, this contribution started from the close connection between individual motivation, self-monitoring, and self-management [26]. Similarly, personal needs and illness experiences often influence women’s participation in an intervention, in accordance with the current literature [27]. This is relevant not only to improving physical features but also at a psychological level (i.e., better self-efficacy and empowerment) [28]. Specifically, this contribution presents five themes of interest as follows: physical well-being, psychological well-being, coping with the illness, social relationships, and support for research.

As presented in the result section, reaching satisfying physical and psychological well-being was the main focus of breast cancer survivors’ interest in participating in this project. Moreover, increasing social relationships (in terms of meeting new people and improving social abilities) was particularly evident. Additionally, women reported the need of promoting strategies to cope with their previous illness experience on a third level. Finally, a few participants highlighted their interest in helping with scientific studies and research. Similar themes were identified as outcomes, but differences in terms of content and relevance between motives and outcomes were observed.

Firstly, physical well-being is at the center of participants’ motives and outcomes. In particular, breast cancer survivors expressed their need to increase their ability to do physical movements as well as relaxation to promote well-being. This aspect is important, especially for breast cancer survivors who could not practice physical activity after their diagnosis [29]. Thereby, in our study, breast cancer survivors appeared prone to physical activities, knowing their efficacy to prevent diseases, such as diabetes and obesity [30], favor the musculoskeletal and cardiovascular system [31], and promote improvements in overall well-being [32]. This is in accordance with changes in body image and self-satisfaction related to physical appearance and attitudes toward the body after breast cancer, also in terms of femininity [33]. Moreover, studies have shown that physical exercise can increase cognitive performance [14,34]. A study by Peterson and colleagues [35] showed improvements in cancer survivors’ memory and executive functions through an aerobic exercise intervention. At the same time, participants showed interest in improving body awareness. Due to oncological treatments, breast cancer survivors have to be aware of inner sensations they have never experienced before, which can affect their body perception [36,37]. This could be helpful to alleviate physical issues [38]. In line with our results, motives and outcomes related to the need to do physical movements, having time to relax, and body awareness are also relevant as outcomes. On the contrary, participants did not report an aesthetic interest after the intervention. This may be related to the focus of psychological interventions on inner sensations and emotions, giving less importance to women’s aesthetic evaluations.

Secondly, in our study, we found that psychological well-being is almost equally relevant. In particular, participants expressed their interest in improving self-acceptance and self-esteem, which are generally damaged by oncological experience [39,40]. Interestingly, motives and outcomes related to psychological well-being do not change before and after intervention. This evidence shows that breast cancer survivors are strongly self-aware about their psychological needs, which may be useful in order to improve tailored and appropriate interventions [41]. Finally, taking time for themselves could be related to the need to take care of oneself, which is important for psychological well-being after a cancer experience in a first-centered and holistic view [42].

Coping with the illness is the third theme of interest that emerged from participants’ motives and outcomes. In particular, breast cancer survivors highlighted the need to cope with the fear of recurrence and negative emotions (e.g., anger and sadness). Current studies show that there is a relationship between fear of cancer recurrence, anxiety, and self-efficacy [43]. Similarly, the literature highlights the importance of coping strategies to enhance illness adjustment [44]. For women with oncological experience, coping is a strategy through which they perceive and handle various stressors that emerge during the breast cancer diagnosis and treatment process [45]. Facing physical and emotional issues related to illness can lead to reframing a new identity as a woman, not only a patient [46]. After breast cancer, people deal with the implications of their changed self-identity, for example, as empowered survivors or women at risk [47,48]. Addressing storylines articulated around illness, autobiographical memories are relevant to promote positive self-representations [5]. This is in line with the results obtained from this intervention, which aimed to distance from the idea of being ill. Accordingly, self-awareness about the illness was promoted by this intervention. Finally, we noticed the relevance of illness acceptance as both a motive and an outcome. Otherwise, the interest in learning new psychological strategies at the beginning disappeared as an outcome. After interventions, participants instead reported the need to be a woman and not only a patient, in line with the psychotherapy aims. This could be interesting in order to analyze a relevant change between the need of “strategies, instruments to cope with illness” and the possibility of promoting well-being through a new identity not only associated with the breast cancer experience.

Moreover, our findings also confirm the relevance of social relationships. Sharing experiences and emotions with people who have experienced a similar situation could be useful [49]. This leads to discussions about sharing experiences with cancer, survivorship and everyday life, and being open to another point of view, as also reported by breast cancer survivors in this project. Group relationships facilitate strategic skill acquisition, encouraging critical reflection about the self and possible changes [50]. Specifically, social relationships increase empathy, understandable emotions and needs, and new friendships, which were also some of the main factors among participants’ take-home benefits. According to the literature, social connections, group support, and peer influence are important parts of an intervention that integrates physical and psychological aspects to promote quality of life [51]. In line with the literature, socio-relational opportunities played a relevant motivational role in participants’ engagement [8]. Accordingly, the need for sharing experience was reported as both a motive and an outcome to promote breast cancer survivors’ well-being, in accordance with the current literature [52]. Otherwise, women highlighted the relevance of improving themselves and their personal habits after interventions. At the same time, their interest in staying in contact with women with similar breast cancer experiences disappeared after the projects. This could be the result of an intervention that leads women to promote their abilities and well-being, starting from their inner resources.

Finally, the interest in supporting scientific research has been reported by some participants. Since there are strong emotional bonds among breast cancer survivors [53], our hypothesis is that participants could perceive the need to be helpful for other people who will live the same experience. Consistently, a study by Kardinal and colleagues [54] highlighted an altruistic reason as a motivation to be involved in a clinical trial. In particular, oncological survivors were motivated to be helpful for medical science advancements for future patients.

### Limitations

The findings of this study add to our understanding of motives and outcomes related to participation in an integrated intervention for breast cancer survivors. The limitations of the present study are presented as follows. Firstly, due to the number of participants, it was not possible to analyze the differences in motives and outcomes between subgroups (e.g., white vs. blue collars). Future studies need to cover this limitation in order to prevent possible sampling bias. Additionally, due to the data collection procedure, another limitation concerns the reports’ shortness. More detailed and longer narrations would have allowed for the adoption of alternative phenomenological approaches, such as interpretative phenomenological analysis [55]. However, the whole corpus under investigation was made up of a good number of reports and was dense in content. This way, it was reliable in order to know participants’ thoughts and emotions, which strengthened our results and allowed interesting insights for future research. For example, future studies with longitudinal approaches could verify if the motives collected change during the project, such as after 2–3 sessions. Second, the results of this study are specifically related to the breast cancer survivor population. The present paper contributed to the understanding of the recent and little-explored motives for participating in a mixed-method program focused on body image issues in this population and adds richness to healthcare professionals’ knowledge. However, each type of cancer is specific and leads to different physical and psychological issues. Future studies might consider whether there are differences in motives and outcomes in reference to a program for people with cancer at a different stage, for example. Third, the nature of the thematic analysis allowed the study of a specific issue or phenomenon in a certain population and in a particular context [56]. This way, some caution should be exercised in generalizing the results. Finally, future research may better explore the motivations of breast cancer survivors in supporting experimental research. The association between defense mechanisms, personality traits, and illness characteristics could be taken into consideration.

## 5. Conclusions

Knowing motives and outcomes related to integrated intervention for breast cancer survivors is relevant to implement useful and tailored programs. Broadening our understanding of the experiences of breast cancer survivors enables healthcare professionals to devise appropriate strategies to provide better care in order to improve their quality of life. In particular, a clear understanding of motives and outcomes would provide a wider range of resources and strategies for coping to tailor interventions. At the same time, professionals with different expertise could work together to tailor an efficacy program. Moreover, a holistic view of physical and psychological needs could be helpful to promote the overall quality of life. Breast cancer survivors should know that cancer does not always exclude the possibility of well-being. Therefore, interventions focused on survivors’ needs could change the way in which cancer survivors deal with their physical and emotional issues. Future research should indeed be designed by taking into account motives and desired outcomes.

## Figures and Tables

**Table 1 behavsci-12-00271-t001:** Sociodemographic characteristics of participants.

	N	%
**Educational level**		
Primary/middle school	18	40%
High school	4	9.1%
Bachelor	21	47.2%
Master’s Degree/Ph.D.	2	3.6%
**Employment**		
Unemployed	1	1.8%
Blue-collar	2	5.5%
White-collar	41	92.7%
**Place of residence**		
North of Italy	34	74.5%
South of Italy	5	10.9%
Island of Italy	6	14.9%

**Table 2 behavsci-12-00271-t002:** Participants’ motives and outcomes related to the intervention.

	Motives	*n*	Outcomes	*n*
**Physical well-being**	*1.1 Doing exercise*: need of physical movements; promotions of an appropriate physical posture	10	*1.1 Reduction of physical issues* (e.g., fatigue, pain, and sleep disturbances)	9
	*1.2 Relaxation*: tension reduction	5	*1.2 Relaxation*: tension reduction	2
	*1.3. Increase physical awareness*	9	*1.3. Increase physical awareness*	10
	*1.4 Aesthetic motives*: appreciate the own body and body image	1		
**Psychological well-being**	*2.1 Self-awareness:* be aware of their inner world to accept themselves and cope negative emotions	17	*2.1 Self-awareness:* be aware of their inner world to accept themselves and cope negative emotions	21
	*2.2 Having time to make new experiences*	13	*2.2 Having time to make new experiences*	5
**Coping with the illness**	*3.1 Illness acceptance:* face with physical and psychological issues related to breast cancer	6	*3.1 Illness acceptance:* face with physical and psychological issues related to breast cancer	10
	*3.2 Learning psychological strategies*	7	*3.2* *“Be a woman and not just a patient”*	5
**Social relationships**	*4.1* *Sharing their illness experience*	12	*4.1* *Sharing their illness experience*	
	*4.2 Stay in contact with people who lived a similar illness experience*	9	*4.2* *Improve yourself and personal habits*	14
	*4.3 previous positive experience in groups*	3		7
**Support to research**	*5.1* *Helping scientific studies*	3	*5.1* *Helping scientific studies*	1

Note: *n* = number of participants involved in each sub-theme.

## Data Availability

The data presented in this study are available on request from the corresponding author due to privacy.

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
