# Peer review of "Breast Cancer Survivors’ Motivation to Participate in a Tailored Physical and Psychological Intervention: A Qualitative Thematic Analysis"

_behavsci, 2022, doi:10.3390/bs12080271_

Round 1

Reviewer 1 Report

Dear editor:

This study explored the differences between the initial motivations and perceived outcomes obtained after an integrated program of both physical and psychological programs to promote well-being in breast cancer survivors. The results are clearly presented, and the manuscript is well structured in most parts. I only have a few comments:

1.       For Table 2, the authors need add two columns to respectively calculate and summarize the number of participants’ who have mentioned the corresponding keywords in “motives” and “outcomes”.

2.       From Table 1, we can see that sampling bias exists among the participants. I think the authors need to analyze the differences of motives/outcomes across different sociodemographic characteristics to see if motives/outcomes are different between subgroups (e.g. white-collar VS blue-collar). In addition, the sampling bias should be considered as one of the limitations in Section 4.1.

3.       [Line 104 - 109] “The sample was constituted as follows: 24 women (Mage = 50.13; SDage =8.00) participated in a group intervention that …… 19 women (Mage = 50.32; SDage = 5.65) participated in a group intervention coupling sailing boat courses with psychological group interventions for a total of 8 sessions”   24+19=43 is not equal to the number of participants the authors mentioned in line 95 (45 patients).

4.       [Line 36] “Fioretti et al., 2017; Sebri et al., 2021a).”  This is an incomplete sentence

Author Response

Dear reviewer,

Thank you for your work on our manuscript and for the positive comments. Starting from your suggestions, we improved this contribution as depicted below. Kindly notice that any modification to the manuscript has been highlighted in green to aid consultation of review, new references included.

  1. As requested, we included two columns in Table 2 in order to calculate and summarize the number of participants who have mentioned the corresponding keywords in “motives” and “outcomes”.
  2. Thank you for noticing this aspect. We included this aspect in the limitation section (Please, see lines 539-540).
  3. Thank you for your comment. We apologize for the mistake and we revised the number of participants. We confirm that the total number of participants is equal to 45 (see lines 94 and 110-115).
  4. Thank you for the comment. We revised the sentence accordingly (see line 35).

Reviewer 2 Report

This is a topic of great relevance and social and health relevance, although the article presents serious methodological problems that must be resolved.

The following changes must be made:

1) The article must adjust in all its extremes to the format of the journal, in its way of citing (with numbers), in its figures, tables, text.

2) The article should be structured in three clear parts: background; methodology and results; discussion and conclusions

3) The antecedents must be restructured and articulated much better. Current empirical evidence must be provided (2022, 2021), and culminate in the research question, specified in the objective and materialized in the hypothesis to be contrasted and demonstrated.

4) The entire Methodology must be reorganized, like the entire article.

--Participants, their composition must be clear: gender x age x levels x others, in a nested way; The use of percentages is not admissible, since it deals with cases and few. The inclusion and exclusion criteria must be provided, the sampling followed, the representativeness of the sample, its generality... The table provided is inadequate, N is missing, it is missing to nest N x gender x age x level and know the real totals, not just percentages

-- The exhaustive and mutually exclusive system of categories used must be provided, indicating its agreement between encoders, its construct validity, its relevance, based on which theoretical model has been developed, based on which accepted constructs have been made. Provide in a table the system of categories, examples, counterexamples of each category with other nearby ones, the coefficients obtained for each category and by groups (frequency, percentage, coefficients, effect size...)

-- Data analysis, should be better clarified, provide the total pool of records obtained, the agreement between observers, software used. If necessary, provide the matrix of records and their significance in supplementary data.

--Design and procedure: the steps followed must be clarified, the logic of the investigation followed; the relevant causal, dependent, mediating variables, if any...

5) The results must follow the steps indicated in Data analysis; provide coherence with respect to what is indicated in the methodology. Clear evidence must be provided in a table, with precise and rigorous analysis; and some illustrative figure of what it contributes

6) Discussion and conclusions. It should be indicated whether or not the research question is answered, its relationship with other current studies (2022, 2021), and with some credible theoretical explanation and with heuristic value. It must be indicated if the objective is achieved or not and why; and whether or not the study hypotheses are met; the limitations of the study must be indicated (reduced sampling, low representativeness, category system that does not allow for replication..., data analysis that is not rigorous...not very reliable...); and his way of solving it; applications for practice and added value of the study or conclusions.

7) The objective of any study is to be able to replicate it and therefore, provide clear, brief and sufficient precise information so that other researchers can verify what they have done. For this reason, you must review the entire article and correct where there is no precise and clear information.

8) All changes must be made in the new version of the article, marked in color, and explained one by one in the reviewer's forum.

In summary, the article presents serious methodological and conceptual problems, which make it recommendable not to publish it. Taking into account the limitation of the sample, the limitation of the analysis carried out, the scant representativeness of the data, it does not seem opportune to consider this article for publication, at least in this version. In the event that the authors do a complete rewrite of the article, with new analyses, with a current and rigorous approach, it should not be considered for publication.

Author Response

Dear reviewer,

Thank you for your time and precious suggestions on our Manuscript. We hope that the revisions improved the quality of the manuscript. Please note that any modifications to the manuscript have been highlighted in green.

Please, see detailed responses below and in the new version of the manuscript.

  1. The article has been adjusted in all its extremes to the format of the journal. We checked and modified all the references, adding also some missing citations, and we revised Tables.
  2. Thank you for your suggestions to promote our contribution. We checked the research manuscript section included in the Journal’s guidelines for authors (https://www.mdpi.com/journal/behavsci/instructions). In this sections, it is recommended to use the following sections of the Manuscript: 1) Introduction; 2) Materials and Methods; 3) Results; 4) Discussion; 5) Conclusion. On this basis, we divided the Manuscript in these five sections. Please, let us know if the Editor and the reviewers suggest to use a different structure.
  3. We would like to thank the reviewer for this comment. However, we think there is a misunderstanding here. The paper presents a qualitative study about the breast cancer survivors’ motivations to participate in an integrative program. This kind of study does not imply the presence of hypotheses to be contrasted and demonstrated. There is extensive literature on this, including the references that are mentioned in the manuscript (e.g., Braun & Clarke 2006; 2022; Clarke & Braun, 2014) and many papers that have been already published in this journal. In order to address this point, we clarified the introduction and the aims of this study in the manuscript.
  4. As requested, we modified the Methodology section as follows:
    1. Thank you for your comment. The participants section has been restructured and some details have been included in the manuscript and in the Tables. We kindly specify to the Reviewer that in qualitative studies the clarification of gender x age x levels x others in a nested way is not generally included (e.g., Oppong Asante, 2019; Swierad et al., 2017; Testoni et al., 2020). Indeed, no statistical analysis was conducted in this study, but data were analyzed in a qualitative way. Inclusion and exclusion criteria were included in the Manuscript (see lines 96-101). Moreover, as reported by Leung (2015, p. 326), most qualitative research studies, if not all, are meant to study a specific issue or phenomenon in a certain population, in a particular context, hence "generalizability of qualitative research findings is usually not an expected attribute".
    2. The analytical process followed the main phases of qualitative thematic analysis, as described by Braun and Clarke (2006). Please, note that in Thematic Analysis the agreement between coders is not expected (Terry & Hayfield, 2021). Moreover, you asked to “Provide in a table the system of categories, examples, counterexamples of each category with other nearby ones, the coefficients obtained for each category and by groups (frequency, percentage, coefficients, effect size...)”. However, quantification in terms of “the coefficients obtained”, “frequency, percentage, coefficients, effect size”, are not consistent with qualitative Thematic Analysis (Braun & Clarke 2022; Terry & Hayfield, 2021). The researchers had several meetings to discuss and reviewed the themes that were identified from the data. We provide additional details on the results in Table 2. In this Table, all the themes and sub-themes related to motives and outcomes are provided. The number of participants involved in each sub-themes was also added to Table 2.
    3. We revised the Data Analysis section. The analytical process followed the main phases of qualitative thematic analysis, as described by Braun and Clarke (2006). In brief, we 1) read each text many times, familiarizing ourselves with the contents; 2) we computed an initial coding of the data; 3) we grouped the different codes into potential main themes (and sub-themes), 4) we reviewed the themes, and 5) we labeled the themes, providing an explicit definition of their contents.. In this qualitative study, no specific qualitative data analysis software (QDAS)was used, but Microsoft Excel was used to support the initial coding procedure.
    4. Following this comment, we clarified the data analysis procedure and we introduced the references to recent step-by-step manuals of Thematic Analysis (Braun & Clarke 2022; Terry & Hayfield, 2021). The agreement between observers was described in terms of how the different researchers’ points of view, were integrated through discussion. We specified the software we used. However, no quantitative index of agreement between the researchers or significance was provided. These are usually features of content analysis, but they are not part of thematic analysis as described by Braun & Clarke in 2006 and by the following researchers in the same line of thoughts. In this regard, we added a reference to the work by Neuerdorf (2018), clarifying the differences between content and thematic analysis.
  5. Data Analysis and Results sections were revised following your suggestions. The analytical process was detailed in the text. Results highlighted the five main themes emerged during data analysis and sub-themes for motives and outcomes.
  6. Thank you for your suggestions. We checked and revised the discussion and conclusion, according to your comment. Moreover, we included new recent references (Ikeda et al., 2020; Kardinal et al., 2020; Savioni et al., 2021; ), as requested. We kindly specificy that we adopted an induptive approach to coding (codes and themes were data driven). Moreover, consistent with our approach and as mentioned above, we did not provide specific hypotheses for this study. Additionally, we added more details in the limitation section, highlighting the possible limitations of the sample characteristics and the need to prevent possible sample bias in future studies. Finally, the conclusion section has been totally revised. The revised version of conclusion includes applications for practice and recommendation for future studies.
  7. We agree with the relevance of replicating studies. This way, we reviewed the entire articles and provided clear information. We added specific information about the intervention on both physical and psychological levels. Particularly, more specific details about characteristics, goals, and benefits of physical exercises and psychological support groups have been provided (see lines 117-133).
  8. Any modifications to the manuscript has been highlighted in green.

References:

Braun, V., & Clarke, V. (2022).Thematic Analysis- A Practical Guide. Sage, London, UK.

Ikeda, M.; Tamai, N.; Kanai, H.; Osaka, M.; Kondo, K.; Yamazaki, T.; ... Kamibeppu, K. Effects of the appearance care program for breast cancer patients receiving chemotherapy: A mixed method study. Cancer Reports, 2020, 3(3), e1242. https://doi.org/10.1002/cnr2.1242

Kardinal, C. G.; Sanders, J. B. Altruism: A form of hope for patients with advanced cancer. Journal of Clinical Oncology. 2010,  28(15_suppl), e19559-e19559.

Leung L. (2015). Validity, reliability, and generalizability in qualitative research. Journal of family medicine and primary care, 4(3), 324–327. https://doi.org/10.4103/2249-4863.161306

Neuendorf, K. A. (2018). Content analysis and thematic analysis. In Advanced research methods for applied psychology (pp. 211-223). Routledge.

Oppong Asante, K. (2019). Factors that promote resilience in homeless children and adolescents in Ghana: A qualitative study. Behavioral Sciences, 9(6), 64.

Savioni, L.; Triberti, S.; Durosini, I.; Sebri, V.; Pravettoni, G. Cancer patients’ participation and commitment to psychological interventions: a scoping review. Psychology & Health. 2021, 1-34. https://doi.org/10.1080/08870446.2021.1916494

Swierad, E. M., Vartanian, L. R., & King, M. (2017). The influence of ethnic and mainstream cultures on African Americans’ health behaviors: a qualitative study. Behavioral Sciences, 7(3), 49.

Terry, G., & Hayfield, N. (2021) Essentials of Thematic Analysis. Washington, DC: American Psychological Association.

Testoni, I., Palazzo, L., De Vincenzo, C., & Wieser, M. A. (2020). Enhancing existential thinking through death education: a qualitative study among high school students. Behavioral Sciences, 10(7), 113.

Reviewer 3 Report

Introduction part.

I would recommend choosing a different word than "journey" (line 30) when talking about the period of the patient's life after the diagnosis. Line 36 begins with the mentioning of sources. What do they refer to? What is meant by the words "their generativity" (line 38)? It doesn't really seem to fit in the text.

Methods.

What is meant by "worked as white-collar employers"? Does it really mean that almost all participants were employers?

How many participants participated? The answers differ – it was mentioned 45, but further on in the text - 24+19, respectively, 43. This needs to be clarified. Therefore, it is not clear for which number of respondents the presented % is. It is recommended to present both the number of respondents and % in the table, and present them separately for each group. It is not clear how the participants were allocated to the intervention groups. Did they choose themselves? It is also unclear why the researchers chose two different interventions, but it does not appear anywhere in the results. What justification/reason for this?

Results.

Given that participants received different interventions, did any differences emerged in the outcomes? In psychological well-being? In social relations?

Discussion.

The desire to support scientific research could be discussed more broadly - the authors believe that it was due to the need to be helpful for others, but one could also look for literature on coping mechanisms or psyche defense mechanisms – altruism, pseudo altruism or reversion. Maybe it was a defense mechanism against their shame?

References.

They should be arranged so that there is a combined entry(?). At this point, the authors have used both “and” and “&”. Surnames are separated by a comma and there is a period after the initial. Also it should be in alphabetical order, for example, the two Sebri et al (35 and 36) have to be changed around. Book sources should be arranged such as 41; 43. Bandura (2) is mentioned in the sources but not cited in the text. Authors' surnames must be written in the same way in both sources and references (6).

Author Response

Dear Reviewer,

Thank you for your relevant comments. We adjusted the Manuscript, as suggested.

Introduction

Firstly, in the Introduction section, we changed the word “journey” with the word “path” that, in a figurative meaning, refers to the way to make progress. Additionally, we specify that the word “generativity” generally refers to the possibility of a pregnancy. As known, oncological treatments and some of the related side effects could impair regular menstruation cycle as well as fertility (Patridge et al., 2008; Sonmezee et al., 2006). Therefore, breast cancer survivors could experience psychological and emotional issues related to changes in their femininity, sexuality, and an overall distortion in their body image (JabÅ‚oÅ„ski et al., 2018; Novaski da Silva et al., 2020; van Oers, 2020; Sebri et al., 2021). We changed this sentence in order to make it clearer.

Methods and Results

Secondly, related to Methods, white-collar employers are defined as “a person who performs professional, desk, managerial, or administrative work. White-collar work may be performed in an office or other administrative setting. White-collar workers include job paths related to government, consulting, academia, accountancy, business and executive management, customer support, design, engineering, market research, finance, human resources, operations research, marketing, public relations, information technology, networking, law, healthcare, architecture, and research and development. Other types of work are those of a grey-collar worker, who has more specialized knowledge than those of a blue-collar worker, whose job requires manual labor” (https://en.wikipedia.org/wiki/White-collar_worker). As reported in other current studies (Azavedo et al., 2021; Kayaba  et al., 2021; Matthews et al., 2021), we considered this definition in order to make a difference between different types of works with the aim of describing the sample in a comprehensive way. It doesn’t mean that all participants are employers. As reported in Table 1, 1.8% of the sample are unemployed. Otherwise, 5.5% of participants are workers involved in manual jobs and 92.7% of the total sample are participants who work in managerial or administrative work. Regarding the sample, we apologize for the error. As reported in the abstract and in Participants section, a total of 45 breast cancer survivors participated in the study. Moreover, we added a new column in Table 1 with the number of respondents related to each percentage. Both the number of respondents and the related percentage are presented in the socio-demographic Table, as requested. Finally, in reference to the method section, this study presents a unique intervention that is focused on physical exercises and psychological sessions in groups. In order to participate, breast cancer survivors take part to a sailing or a physical exercises course (chosen by the participants) with the aim of promoting physical well-being; moreover, psychological sessions were conducted aiming at decreasing emotional distress related to illness experience and promoting positive behaviors. Accordingly, results are referred to a unique sample because of a unique intervention group. We added this information at lines 117-133).

Discussion

Regarding the discussion, current literature shows that altruism is linked to a sense of connection to society broadly, science, and community partners (Carrera et al., 2018). For breast cancer survivors, the feeling of being part of a group with people who had a similar oncological experience (on both a physical and psychological levels) could reinforce social connections and the sense of being part of a community. This is in line with the study by Kardinal and colleagues (2019) in which 38% of patients cited altruistic reasons for participation in a clinical trial.  Although the main motivation to participate was related to the hope of therapeutic benefits, the patients’ engagement in the advancement of medical science to help future patients is an interesting result (Kardinal et al., 2010). Aiming at promoting the discussion about supporting scientific research, we added these considerations at lines 517-520. At the same time, we argue that this point could be better explored in future studies, as suggested in the limitation section.

References

Finally, references have been checked and modified into the text as well as in the reference list by following the author guidelines.

References:

Azevedo, L. M., Chiavegato, L. D., Carvalho, C. R., Braz, J. R., Nunes Cabral, C. M., & Padula, R. S. (2021). Are blue-collar workers more physically active than white-collar at work?. Archives of Environmental & Occupational Health, 76(6), 338-347.

Carrera, J. S., Brown, P., Brody, J. G., & Morello-Frosch, R. (2018). Research altruism as motivation for participation in community-centered environmental health research. Social science & medicine, 196, 175-181.

JabÅ‚oÅ„ski, M. J., Streb, J., Mirucka, B., SÅ‚owik, A. J., & Jach, R. (2018). The relationship between surgical treatment (mastectomy vs. breast conserving treatment) and body acceptance, manifesting femininity and experiencing an intimate relation with a partner in breast cancer patients. Psychiatr. Pol, 52(5), 859-872.

Kardinal, C. G., & Sanders, J. B. (2010). Altruism: A form of hope for patients with advanced cancer. Journal of Clinical Oncology, 28(15_suppl), e19559-e19559.

Kayaba, M., Sasai-Sakuma, T., Takaesu, Y., & Inoue, Y. (2021). The relationship between insomnia symptoms and work productivity among blue-collar and white-collar Japanese workers engaged in construction/civil engineering work: a cross-sectional study. BMC public health, 21(1), 1-8.

Matthews, L. R., Gerald, J., & Jessup, G. M. (2021). Exploring men’s use of mental health support offered by an Australian Employee Assistance Program (EAP): Perspectives from a focus-group study with males working in blue-and white-collar industries. International journal of mental health systems, 15(1), 1-17.

Novaski da Silva, F. C., Arboit, É. L., & Possamai Menezes, L. (2020). COUNSELING OF WOMEN THROUGH ONCOLOGICAL TREATMENT AND MASTECTOMY AS A REPERCUSSION FROM BREAST CANCER. Revista de Pesquisa: Cuidado e Fundamental, 12(1).

Partridge, A. H., Gelber, S., Peppercorn, J., Ginsburg, E., Sampson, E., Rosenberg, R., ... & Winer, E. P. (2008). Fertility and menopausal outcomes in young breast cancer survivors. Clinical breast cancer, 8(1), 65-69.

Sonmezer, M., & Oktay, K. (2006). Fertility preservation in young women undergoing breast cancer therapy. The oncologist, 11(5), 422-434.

van Oers, H. (2020). Body image and the psychological and behavioural indices of distress in female breast cancer patients. World Scientific News, 140, 172-184.

Round 2

Reviewer 2 Report

The improvement work carried out by the authors is appreciated.

However, the fundamental problems indicated in the first review remain:

1) The central idea of ​​any investigation is to be able to be replicated by other researchers and to be able to contrast and verify that, based on the data or evidence or documents or observations, it is possible to verify and repeat similar results. If everything is interpretation, it is not possible to confirm it, and therefore we proceed to a reflection of the authors, but not an investigation based on evidence and data.

2) The authors indicate that, as it is a qualitative study, it is not possible to ask a research question or problem under study, nor make hypotheses or forecasts. So, what is it that is intended to be solved with this research -and I am not referring to the objective, but to the scientific problem or research question?

3) They also state that it is not possible to nest the data in table 1 (N=45) by gender x age x characteristics; So, how is it possible to verify that what they affirm has some value or some generality?

4) The basic instrument of a qualitative methodology is to use a system of exhaustive and mutually exclusive categories (they are called codes and themes); but they have to be clearly defined and know what they are, to be able to replicate it and to be able to verify that the evidence they provide is generalizable and goes beyond pure reflections and interpretations without evidence.

5) The control of the quality of the process, in the qualitative methodology is even more flexible than in the methodology that you call quantitative; How to guarantee that the classifications in codes and topics are appropriate, reliable and valid, generalizable, have some added value and based on what evidence in data (observations, responses, documents, etc.)?

6) The explanation of the analysis of the evidence and the presentation of results does not allow replicating the results and we cannot know if it is the authors' opinions based on their reflections and interpretations (more or less correct) and not on the empirical evidence?

7) For all these reasons, I consider that this work is not suitable or ready for publication in a journal of the category and quality of Behavioral Sciences; if you want the journal to maintain an international level of rigor, quality, replicability, contrast of empirical evidence; The quality of the work must be kept high. In this case, at least if the authors responded to the questions and changes indicated in the first revision, it would be admissible to consider some possibility of recommending its publication; under these conditions, I can only recommend its non-acceptance

Author Response

Dear reviewer, thank you for your appreciation. It is unfortunate that you believe that the “fundamental problems” remain. Even if a long dissertation about the assumptions of qualitative methods probably would not be appropriate here, we did our utmost to provide a point-by-point answer to your comments.

1) We are aware that many researchers adopt the reviewer’s perspective in producing knowledge, as we often do in our daily practice. However, this study is based on a qualitative approach and qualitative research has specific characteristics that should be considered. In this regard, a number of internationally recognized criteria have been published. Just to mention one publication with a specific focus on thematic analysis, we suggest the recent paper by Braun and Clarke (2021). As you will see, most of the issues that the reviewer raised are not consistent with the mentioned criteria. Further details are present in the following answers.

2) In simple terms, the thematic analysis seeks to describe patterns across qualitative data. One of its definitions is an “interpretative approach to qualitative data analysis that facilitates the identification and analysis of patterns or themes in a given data set (Braun and Clarke 2012)”. We never stated that “it is not possible to ask a research question or problem”, but there are no hypotheses to be tested. The point is that in this study it was possible to describe and reflect on the patients’ motivations and perceived outcomes, so the main research question was “what are the patients’ motives and the perceived outcomes associated with the intervention?”. The method is consistent with this research question. For more examples of research questions that can be answered with thematic analysis, it is possible to consult a large number of published papers or visit the website: https://www.thematicanalysis.net/understanding-ta/

3) We never stated that “it is not possible to nest the data in table 1”. In the previous round of revisions, we clarified that “The participants' section has been restructured and some details have been included in the manuscript and in the Tables”. We kindly specified that in qualitative studies the clarification of gender x age x levels x others in a nested way is not generally included (see the previously provided references of published papers in this journal). In any case, we are at your disposal to include the table if you and the Editor agree with this integration.

4) As stated by Anderson (2017), statistical generalizability is not desirable or reasonable in qualitative research but qualities of ‘thick description’ that present findings with categories identified that are appropriately defined and supported by sufficient data are expected. The criterion of  “transferability” can be thus achieved through the presentation of rich, direct quotations or the authors’ own words, descriptive phrases, or experiences that convey a sense of the participants and their environment as the basis for careful interpretation to illustrate in-depth concepts and constructs that are important to the study. 

On the other hand, reflexivity is very important (see: Terry & Hayfield, 2021). Reflexivity is defined as “researchers’ critical self-awareness: the process by which they examine the understanding of self/other and analyze the ways in which their preconceptions influence and impact the research” (Finaly 2016; 2021). A ‘good’ reflexivity allows to make the research process more transparent (Finlay, 2021). Indeed, the term “reflexive” does not only demarcate it as a particular thematic analysis approach, but it emphasizes the importance of the researcher’s subjectivity as an analytic resource, and their reflexive engagement with theory, data and interpretation (Braun & Clarke, 2021). Accordingly, a Thematic Analysis thus interrogates and makes transparent the researcher’s role in the production of knowledge regarding the topic of interest. This way, it is not about following procedures to ensure inter-reliability/consensus. Braun and Clarke specified that researchers have to be thoughtfully  and  reflexively  engaged  with the data and the process (see more details in this comprehensive summary of their method on the University of Auckland website: https://www.psych.auckland.ac.nz/en/about/thematic-analysis.html)

5) We agree with the reviewer that the control of the quality of the process is relevant in every kind of research. In this regard, see also the answer we provided to point 1. We are not saying that our work is without limits (and we illustrate the limitations of this study in the discussion section), but the criteria that this specific reviewer used to evaluate our manuscript, from the first round, were not appropriate.

6) We believe that this comment is related to the concept of replicability. The reviewer’s ideas are not new and this point has been largely debated in qualitative research and is still debated. By the 1960s, qualitative research was attacked by US quantitative sociologists who viewed qualitative inquiry as lacking objectivity, validity, reliability, and replicability (Bryant & Charmaz 2007). Even if around sixty years have passed, concerns about objectivity, validity, reliability and replicability in qualitative research still pervade academic psychology. For example, Anczyk et al. (2019) plea for replication in qualitative research. Rubin, Bell, and McCleland (2018) document the above concerns in a mixed methods study of graduate psychology programs.

However, the debate is still open. In some of the main approaches in qualitative research, like grounded theory, interpretative phenomenological analysis, and ethnography - and in many of their variations - the authors prefer not to adopt the criterium of replicability, but to use other criteria (for discussions on ‘quality’ in these approaches see Charmaz & Thornber, 2022; Nizza et al., 2021). The authors provide many reasons for this, on the one hand rejecting the importance of the traditional positivist criteria, and on the other proposing different ones (...). The method that we used in our study, (reflexive) thematic analysis thus shares with its “cousins” a similar perspective about the use of “replicability”. 

7) We understand your position and we are also interested in helping the journal Behavioral Sciences in keeping a high level. For this reason, in the previous round of revisions and in the present one, we answered all the questions, supporting our answers with the scientific literature. Just to mention one example, the methodological paper by Braun & Clarke (2006) has more than 170.000 citations in Google Scholar (>63.000 in Scopus). Some of the reviewer’s personal opinions are not in line with the core of that methodological paper and with the other references (both papers and manuals) that we provided in the main manuscript and in the previous cover letter.

We conclude this answer by saying that in the previous round of revisions we addressed a number of comments, encountering the other reviewers and the Editor’s approval. In any case, we appreciate the request by Miss Ioana Serban - Assistant Editor, who gave us the possibility to reconsider our manuscript and to further explain our position. We thank the reviewer for the time spent reading our work and we apologize for this long cover letter.

Best regards

References

Anczyk, A., H. Grzymała-Moszczyńska, A. Krzysztof-Świderska, and J. Prusak. 2019. The

replication crisis and qualitative research in the psychology of religion. The International. Journal for the Psychology of Religion 29:278–91. http://doi.org/10.1080/10508619.2019.1687197

Anderson, V. (2017). Criteria for evaluating qualitative research. Human Resource Development Quarterly, 1-9.

Braun, V., & Clarke, V. (2006). Using thematic analysis in psychology. Qualitative research in psychology, 3(2), 77-101. https://doi.org/10.1191/1478088706qp063oa

Braun, V., & Clarke, V. (2012). Thematic analysis. In: Cooper, H., Camic, P.M., Long, D.L., Panter, A.T., Rindskopf, D., Sher, K.J. (eds.) APA Handbook of Research Methods in Psychology, Research Designs, vol. 2, pp. 57–71. American Psychological Association, Washington 

Braun, V., & Clarke, V. (2021). One size fits all? What counts as quality practice in (reflexive) thematic analysis? Qualitative Research in Psychology, 18(3), 328–352. https://doi.org/10.1080/14780887.2020.1769238

Bryant, A., and K. Charmaz 2007. Grounded theory in historical perspective: An epistemological

account. In The Sage handbook of grounded theory, A Bryant and K Charmaz (Eds.), pp.31–57. London: Sage.

Charmaz, K., & Thornberg, R. (2021). The pursuit of quality in grounded theory. Qualitative research in psychology, 18(3), 305-327. http://doi.org/10.1080/14780887.2020.1780357

Finlay, L. (2006). ‘Rigour’,‘ethical integrity’or ‘artistry’? Reflexively reviewing criteria for evaluating qualitative research. British Journal of Occupational Therapy, 69(7), 319-326. https://doi.org/10.1177/030802260606900704

Finlay, L. (2016). Championing “reflexivities”. Qualitative Psychology, 4(2), 120-125. http://doi.org/10.1037/qup0000075

Finlay, L. (2021). Thematic Analysis:: The ‘Good’, the ‘Bad’and the ‘Ugly’. European Journal for Qualitative Research in Psychotherapy, 11, 103-116.

Nizza, I. E., Farr, J., & Smith, J. A. (2021). Achieving excellence in interpretative phenomenological analysis (IPA): Four markers of high quality. Qualitative Research in Psychology, 18(3), 369-386. http://doi.org/10.1080/14780887.2020.1854404

Rubin, J. D., S. Bell, and S. I. McCleland. 2018. Graduate education in psychology: Current trends and recommendations for the future. Qualitative Research in Psychology 15 (1):29–50. http://doi.org/10.1080/14780887.2017.1392668.

Terry, G., & Hayfield, N. (2021) Essentials of Thematic Analysis. Washington, DC: American Psychological Association.

https://www.psych.auckland.ac.nz/en/about/thematic-analysis.html

https://www.thematicanalysis.net/understanding-ta/

Round 3

Reviewer 2 Report

While the authors have made some change of interest. The basic problems remain:

1) The background of the study on which they are based should be current (2022, 2021); to provide empirical evidence to substantiate and compare the results obtained.

2) All research, including any review, must contain the research question, the objective and the forecasts that are expected or hypotheses.

3) Table 1 is still a list, not a nested cross table. How many participants are employed and what is their level of education, for example? The table must have totals of all the possible crossings or nesting, to obtain an intuitive vision of how it was shown to be able to draw conclusions

4) The explanation of the Data Analysis has improved compared to the previous version, but it is still totally insufficient. How can another researcher replicate your study and be able to draw the same conclusions?

5) The methodology followed cannot be replicated, the system of exhaustive and mutually exclusive categories should be provided that allow data to be obtained, agreement between encoders, examples and counterexamples, and it is simply about linear categories that can overlap with each other with what which the result of your application depends only on each applicator

6) They should provide more depth, solidity, articulation and comparison with current empirical studies (2022, 2021), also in Discussion and conclusions

7) Discussion and conclusions: is the initial research question answered, is the objective achieved, are the forecasts confirmed? Compared to other recent international empirical reference studies (2022, 2021), how do they differ and in what way? what coincide, what theoretical interpretation is made. In relation to other similar recent studies (which should have been reviewed in the background precisely indicating the methodology followed: 2022, 2021), were comparable results found or is there no study comparable to yours? What were the limitations of the study and the ways to overcome it? What are the applications for practice? What is the added value of the study, or conclusions, based on the data and not based on personal opinions?

8) In general, there is a lack of depth, methodological rigor (it is not possible to replicate the study beyond providing personal opinions that have little to do with the data, without using a powerful instrument in qualitative methodology such as providing a system of exhaustive categories and mutually exclusive that serves as a basis to be able to extract other records and data from other samples in comparable situations The quality controls used are unknown: agreement between observers and how many, evidence extracted from the instrument used (exhaustive and mutually exclusive category system), analysis of the data purely descriptive and enumerative instead of articulated and in depth.

For all these reasons, I recommend its non-publication.

Author Response

Dear Reviewer,

We will like to thank you for your time and the suggestions to review this Manuscript.

We included in the limitation section some criticisms related to the Thematic Analysis as a matter of fact, as suggested by the Academic Editor.